# Exportin 1 (XPO1) Expression and Effectiveness of XPO1 Inhibitor Against Canine Lymphoma Cell Lines

**DOI:** 10.3390/vetsci12080700

**Published:** 2025-07-26

**Authors:** Hardany Primarizky, Satoshi Kambayashi, Kenji Baba, Kenji Tani, Masaru Okuda

**Affiliations:** 1Laboratory of Veterinary Internal Medicine, Joint Graduate School of Veterinary Medicine, Yamaguchi University, 1677-1 Yoshida, Yamaguchi 753-8515, Japan; hardany-p@fkh.unair.ac.id (H.P.); kbaba@yamaguchi-u.ac.jp (K.B.); 2Veterinary Clinical Division, Department of Veterinary Science, Faculty of Veterinary Medicine, Universitas Airlangga, Campus C Mulyorejo, Surabaya 60115, Indonesia; 3Laboratory of Veterinary Internal Medicine, Joint Faculty of Veterinary Medicine, Yamaguchi University, 1677-1 Yoshida, Yamaguchi 753-8515, Japan; 4Laboratory of Veterinary Surgery, Joint Faculty of Veterinary Medicine, Yamaguchi University, 1677-1 Yoshida, Yamaguchi 753-8515, Japan; ktani@yamaguchi-u.ac.jp; 5Laboratory of Veterinary Internal Medicine, Graduate School of Agricultural and Life Sciences, The University of Tokyo, 1-1-1 Yayoi, Bunkyo-ku, Tokyo 113-8657, Japan; okudamu@g.ecc.u-tokyo.ac.jp

**Keywords:** canine lymphoma, cell lines, dogs, exportin 1, immunomodulation, in vitro study, protein transport, XPO1 inhibitor

## Abstract

Lymphoma is one of the most common types of blood malignancies in dogs. Cancer cells often grow because a protein called Exportin 1 moves important protective proteins out of the cell’s control center (the nucleus), thereby abrogating their function. Exportin 1 inhibitors disrupt the translocation and restore their function to inhibit cancer growth, stop the cell cycle, and cause cell death. This study investigated Exportin 1 in canine lymphoma cell lines. When treated with one of the Exportin 1 inhibitors, all tested cell lines significantly reduced their growth and decreased survival. While there was a correlation between the expression of Exportin 1 gene and protein, no clear correlation was found between the gene/protein levels and drug sensitivity. Nonetheless, the results in this study suggest that Exportin 1 could be a promising target for treating lymphoma in dogs.

## 1. Introduction

Lymphoma is a common and considerably aggressive hematologic malignancy that affects the lymphatic system, which is crucial to immune function [1,2]. The similarities between canine and human lymphoma include cellular morphology, clinical manifestations, pathology, tumor behavior, molecular biology investigations, genetic aberrations, diagnostic methods, and therapeutic approaches [2,3,4]. Canine lymphoma is characterized by the uncontrolled proliferation of lymphocytes, with multicentric lymphoma being the most prevalent, which leads to the formation of tumors in lymph nodes, spleen, liver, and other organs and presents with symptoms such as generalized lymphadenopathy, anorexia, loss of activity, and organ dysfunction [1,4].

In canine lymphoma, systemic treatments, particularly multi-drug chemotherapy, are the most effective options, achieving sub-optimal response rates. Chemotherapy has significantly improved quality of life and longevity in many cases; however, most reports resulted in relapse and/or resistance to chemotherapy [1,5,6]. Numerous studies focusing on targeted therapies and immunotherapies emerged, aiming to overcome the challenge of cancer treatments effectively [7,8]. The increasing number of comparative studies in various works of cancer research aims to enhance therapy efficiency, support current chemotherapy agents’ work, and potentially substitute previous treatments with minimal side effects [3,4,6]. Hence, there is an urgent need to identify novel therapy targets to improve the adverse outcomes of canine lymphoma patients.

In human oncology, the nuclear export receptor Exportin 1 (XPO1), also called Chromosome Region Maintenance 1 (CRM1), has gained attention as a promising therapeutic target due to its crucial role in transporting various proteins and RNA molecules from the nucleus to the cytoplasm, thereby maintaining cellular homeostasis and regulating key signaling pathways [9,10]. XPO1 overexpression in cancer cells can lead to the mislocalization of multiple tumor-suppressing proteins (TSPs) including p21, p53, p73, p27, Forkhead box O (FOXO), Retinoblastoma (Rb)1, Adenomatous polyposis coli (APC), Breakpoint Cluster Region-Abelson (BCR-ABL), Inhibitor of kappa B (IκB), and Protein Phosphatase (PP)2A, and contribute to oncogenesis and cancer progression [7,11,12].

Recent studies have highlighted the potential of XPO1 inhibitors in targeting these pathways to induce apoptosis and inhibit tumor growth. Selective inhibitors of nuclear export (SINE) compounds, such as KPT-335 (verdinexor), have been developed to target XPO1 [13,14,15,16]. A phase I study demonstrated that KPT-335 is safe and exhibits biological activity in dogs with spontaneous cancers [5]. The study also revealed potent activity of KPT-335 in one cell line each for canine diffuse large B-cell lymphoma (DLBCL), mast cell tumor, melanoma, and osteosarcoma, as well as clinical samples of canine DLBCL cultured with CD40L stimulation. However, the factors determining susceptibility to KPT-335 remain unclear.

Further, a phase II study involving 58 dogs with naïve or relapsed B-cell and T-cell lymphoma reported that oral administration of KPT-335 was well tolerated, with mild anorexia being the most common side effect [17]. Among the evaluable dogs, the objective response rate (ORR) was 37% (20/54), with approximately 30% in B-cell lymphomas and about 70% in T-cell lymphomas. Given that canine T-cell lymphomas typically have a poor prognosis and respond less effectively to conventional treatments compared to B-cell lymphomas [1,3,6], the observed efficacy of KPT-335 in T-cell lymphoma suggests its potential therapeutic utility. Therefore, this in vitro study aims to investigate whether the sensitivity to KPT-335 is related to *XPO1* gene and/or protein expression, and investigated potential differences in drug sensitivity between B-cell and T-cell lymphoma cell lines. First, we identified whether the XPO1 mRNA and protein levels were overexpressed and have a correlation. Then, we examined if the effectivity of the XPO1 inhibitor (KPT-335) was related to the *XPO1* gene and protein expression in canine lymphoma cell lines. In the context of veterinary oncology, this work will provide auspicious implications for future research and clinical applications in canine lymphoma.

## 2. Materials and Methods

### 2.1. Cell Lines

A total of eight established canine lymphoma cell lines were used in this study. B-cell lymphoma cell lines include 17-71, CLBL-1, and GL-1, whereas T-cell lymphoma cell lines are CLC, CLGL-90, Ema, Nody-1, and UL-1 (Appendix A) [18]. A western blot analysis used a human lymphoblastoid cell line derived from Burkitt’s lymphoma patient, Raji, as a positive control for XPO1 protein expression [8]. Live cells with a 90% or above proportion were eligible to start the cell culture process. The R10 complete medium [RPMI-1640 (FUJIFILM Wako Pure Chemical Corporation, Tokyo, Japan) supplemented with 10% fetal bovine serum (FBS), 1% penicillin-streptomycin (Nacalai Tesque, Kyoto, Japan), and 55 µM 2-mercaptoethanol] was used to sustain all canine lymphoma cell lines. All cells were cultivated at 37 °C in a humidified incubator with 5% CO_2_. Peripheral blood mononuclear cells (PBMCs) were collected from healthy Beagle dogs that were kept for blood donation by Yamaguchi University Animal Medical Center (YUAMEC) Japan, using a density-gradient centrifugation method, and these were used as normal blood cells mainly containing lymphocytes and monocytes.

### 2.2. Quantitative Real-Time Polymerase Chain Reaction (qRT-PCR)

Total RNA was isolated from the canine lymphoma cell lines using a NucleoSpin^®^ RNA kit (Macherey-Nagel, Düren, Germany) according to manufacturer’s instructions. Complementary DNA (cDNA) was synthesized from up to 0.5 μg of total RNA using ReverTra^®^ Ace qPCR RT Master Mix with gDNA Remover kit (TOYOBO, Osaka, Japan). The NanoDrop™ 2000 Spectrophotometer (Thermo Fisher Scientific, Waltham, MA, USA) was used to measure the total quantity of RNA and cDNA. The absorbance waveform was utilized to determine the purity of nucleic acid. The RNA and cDNA were frozen at −80 and −30 °C for later use, respectively. The primers utilized for amplification were as follows: ribosomal protein L32 (*RPL32*), forward primer (5′-TGGTTACAGGAGCAACAAGAA-3′), and *RPL32* reverse primer (5′-GCACATCAGCAGCACTTCA-3′) [19]; *XPO1* forward primer (5′-TGTGACAGGGCTTTTCAGCT-3′) and *XPO1* reverse primer (5′-CCTGTCGAAGGGCTGTTTCT-3′) were specifically designed based on canine sequence (XM_038680688). According to the manufacturer’s protocol, the cells’ cDNA was mixed with *XPO1* or *RPL32* primers using THUNDERBIRD^®^ Next SYBR^®^ qPCR Mix (TOYOBO, Osaka, Japan). The qRT-PCR was performed to analyze mRNA expression levels using the CFX96 Touch Real-Time PCR Detection System (Bio-Rad, Hercules, CA, USA) with the thermal cycler condition as follows: 95 °C for 30 s (initial denaturation); 95 °C for 5 s, 60 °C for 10 s, and 72 °C for 10 s for 40 cycles (thermal cycling); and then performing melting curve analysis.

Expression levels were normalized to *RPL32*, which served as the endogenous control. *RPL32* was stably expressed in our sample sets and accurate for normalization in canine gene expression studies [19]. All reactions were performed in triplicate including no-template controls for each gene and evaluated in three independent experiments. Quantifications of *XPO1* gene expression for all real-time PCR data were calculated using the comparative threshold cycle method (Ct). The ΔCt was calculated by deducting the *XPO1* CT value from the *RPL32* CT value. The expression levels of XPO1 mRNA were determined as 2^−ΔCt^.

### 2.3. Western Blotting

All canine lymphoma cell lines were selected to assess the presence of XPO1 proteins. The whole cell lysates were lysed with NP40 lysis buffer [50 mM Tris HCl (pH 7.5), 150 mM NaCl, 1 mM EDTA, 1% NP40] mixed with phosphatase and protease inhibitors (Thermo Fisher Scientific, Waltham, MA, USA). Proteins were subjected to BCA protein assay (Takara Bio, Shiga, Japan) to determine the protein concentration. Then, each protein sample was loaded into lanes in 6% or 12% SDS-polyacrylamide gel electrophoresis (PAGE) gel to detect the XPO1 (6% gel) and β-actin (12% gel) protein.

Following the electrophoresis, proteins were transferred to polyvinylidene difluoride (PVDF) membranes (Merck Millipore, Billerica, MA, USA) and blocked with 5% skimmed milk mixed with Tris-buffered saline containing 0.1% Tween-20 (TBST) before probing with mouse monoclonal anti-XPO1 (dilution 1:2000; C-1; Santa Cruz Biotechnology, Dallas, TX, USA) overnight at 4 °C on a rotary plate. To obtain an endogenous control, a mouse monoclonal anti-β-actin (dilution 1:5000; AC-15; Sigma-Aldrich, Saint Louis, MO, USA) diluted in 0.5% skimmed milk-TBST was used. After being washed three times with TBST, the membranes were probed with secondary labeling using goat anti-mouse IgG HRP (dilution 1:4000; Santa Cruz Biotechnology, Dallas, TX, USA) and incubated for one hour at room temperature on a rotary plate. At the end of the incubation period, they were re-washed three times with TBST, then incubated for 5 min with SuperSignal™ West Pico PLUS Chemiluminescent Substrate reagent (Thermo Fisher Scientific, Rockford, IL, USA) and visualized using an AMERSHAM ImageQuant 800 (GE Healthcare Bio-Sciences AB, Uppsala, Sweden). The quantification of the signal intensity of XPO1 protein expression was analyzed using ImageJ software version 1.54d (National Institutes of Health, Bethesda, MD, USA) and normalized with the intensity of β-actin.

### 2.4. Cell Proliferation Assay

To examine half-maximal inhibitory concentration (IC_50_) of KPT-335, a Water-Soluble Tetrazolium salt (WST) assay was conducted using the cell counting kit-8 assay (CCK-8; Dojindo Laboratories, Kumamoto, Japan). KPT-335 (Selleck Chemicals, Houston, USA), was dissolved in DMSO at a concentration of 50 mM. Eight canine lymphoma cell lines at a density of 2 × 10^5^ to 8 × 10^5^ cells/mL were treated with KPT-335 at various concentrations (0, 0.001, 0.01, 0.1, 1, 10, and 100 μM) in 96-well plates (triplicate) for 48 h. Cell proliferation was assessed by adding 10 μL WST-8 (2-[2-methoxy-4-nitrophenyl]-3-[4-nitrophenyl]-5-[2,4-disulfophenyl]-2H-tetrazolium, monosodium salt) and incubated for 4 h. The absorbance at 450 nm wavelength was measured using a Thermo Scientific™ Multiskan™ FC microplate reader. The cell proliferation was calculated using the ratio of the absorbance of treated cells to the absorbance of control cells, and the data were expressed as percentages. The IC_50_ of KPT-335 was calculated using Excel software (Microsoft) after plotting proliferation values on a logarithmic curve.

### 2.5. Trypan Blue Exclusion Assay

The canine lymphoma cell lines were seeded at densities ranging from 1 × 10^4^ to 4 × 10^4^ cells in 96-well plates. All groups of cells were treated with either 0.2% of DMSO, 0.125, 0.25, or 0.5 μM of KPT-335 for 48 h. Each cell line was seeded in triplicate for each treatment group and the entire experiment was repeated three times. The KPT-335 treated cells were harvested after 48 h of drug treatment. The trypan blue exclusion assay was performed to determine the number of viable cells present in a cell suspension. The cells were stained with trypan blue after 48 h drug treatment and viable cells were counted with a hemocytometer. The percentage of viable cells was calculated with the total cell count.

### 2.6. Statistical Analyses

All statistical analyses were conducted by JMP^®^ Pro software version 18.1.1 (SAS Institute, Tokyo, Japan). XPO1 mRNA and protein expression levels were analyzed using one-way ANOVA and the non-parametric Kruskal–Wallis test, respectively. Significance values were compared with the control using Dunnett’s method. One-way ANOVA followed by Dunnett’s method for comparisons with control was used for WST and trypan blue assay to compare with no KPT treatment cells. The *p*-value < 0.05 was considered significant.

## 3. Results

### 3.1. Overexpression of the XPO1 mRNA in Canine Lymphoma Cell Lines

We evaluated the *XPO1* gene expression in the eight canine lymphoma cell lines, including three B-cell lines (17-71, CLBL-1, and GL-1) and five T-cell lines (CLC, CLGL-90, Ema, Nody-1, and UL-1) as well as PBMCs obtained from a healthy dog. The qRT-PCR analysis revealed that 17-71 and CLBL-1 from B-cell lines, as well as CLC, CLGL-90, and UL-1 from T-cell lines, showed a significant increase in XPO1 mRNA expression when compared to that of canine PBMCs as a normal cell control (Figure 1a). CLC and UL-1 from T-cell lines showed a significant increase in relative expression to the *RPL32* gene with values of 0.1538 ± 0.033 and 0.0810 ± 0.033, respectively, and *p*-values < 0.01. Meanwhile, significant increases in relative expression of 17-71 and CLBL-1 from B cells and CLGL-90 from T cells were observed in sequence at 0.0520 ± 0.021, 0.0513 ± 0.013, and 0.0562 ± 0.014, respectively, with *p*-values < 0.05. In contrast, there was no significant difference between GL-1, Ema, or Nody-1 cell lines and the PBMCs. In total, five of eight cell lines showed significant overexpression of the XPO1 mRNA.

### 3.2. Overexpression of XPO1 Protein in the Canine Lymphoma Cell Lines

Western blot analysis was conducted to investigate the expression of the XPO1 protein in the canine lymphoma cell lines and the PBMCs as normal control, as well as a Raji cell line as a positive control. The expression of XPO1 protein in the PBMCs was less expressed when compared to those of the canine lymphoma cell lines. In the B-cell lines, only 17-71 showed significant higher XPO1 protein expression with a *p*-value < 0.05 compared to those of CLBL-1 and GL-1. In the T cells, the expression of XPO1 protein in CLC and UL-1 was significantly high with *p*-values < 0.01, and CLGL-90 showed a significant expression with a *p*-value < 0.05. CLBL-1, GL-1, Ema, and Nody-1 had lower XPO1 protein expression compared to the other cell lines, and showed no significant difference compared to that of the PBMCs (Figure 1b,c).

Based on the expression results of the *XPO1* gene and protein, it was summarized that *XPO1* gene expressions in eight canine lymphoma cell lines seemed to be generally higher than those in normal cells (PBMCs) and related to their XPO1 protein expressions, except for CLBL-1, where the XPO1 protein expression analysis yielded no significant difference. However, specific trends in neither XPO1 mRNA nor protein expressions were observed in B-cell lines and T-cell lines.

### 3.3. KPT-335 Inhibits the Proliferation of Canine Lymphoma Cells

To examine an inhibitory effect of KPT-335, an XPO1 inhibitor, a WST assay was conducted using various concentrations of KPT-335 ranging from 0 to 100 μM, and the IC_50_ values of KPT-335 were calculated from eight canine lymphoma cell lines. As shown in Figure 2, there was a considerable reduction in cell proliferation depending on the concentration of KPT-335 with IC_50_ values ranging at the nanomolar level between 89.8 and 418 nM in B- and T-cell lines.

In the B-cells group, KPT-335 had IC_50_ values of 89.8 ± 2.07, 108.6 ± 30.54, and 294.3 ± 25.57 (nM) for 17-71, CLBL-1, and GL-1, respectively. Meanwhile, in the T-cells group, CLC, CLGL-90, Ema, Nody-1, and UL-1 had IC_50_ values of 224.7 ± 78.38, 215 ± 50.96, 147.8 ± 49.69, 220.5 ± 27.34, and 418 ± 78.6 (μM), respectively (Table 1).

Next, we examined a cell viability to further assess the effect of KPT-335 on the canine lymphoma cell lines. The cell viability measured by a cell counting with trypan blue exclusion revealed a decrease in the number of viable canine lymphoma cells (Figure 3). Exposure to 0.125, 0.25, and 0.5 μM of KPT-335 reduced the cell viability significantly in all eight canine lymphoma cell lines compared to non-treated cells (0 μM). This finding was consistent with the results of the cell proliferation assay, which showed that KPT-335 inhibited cell proliferation in a dose-dependent manner. All the cell lines used in this study demonstrated a considerable decrease in the average percentage of living cells. At 0.5 μM concentration, all canine lymphoma cell lines exhibited a percentage of viable cells less than 50%.

## 4. Discussion

In cancers, the mislocalization of many tumor-suppressing proteins (TSPs) caused by the overexpression of the nuclear export protein XPO1 contributed to cell proliferation and increased metastatic potential [9,20]. Canine cancer studies reported the overexpression of XPO1 mRNA levels from patient-derived tissues and cell lines in osteosarcoma [14]. Notably, while XPO1 protein expression has been previously reported in canine DLBCL cells, including the CLBL-1 cell line and cryopreserved tumor cells [5], as far as we know, this study is the first to report XPO1 mRNA expression in canine lymphoma cells.

The expression levels of XPO1 mRNA and protein were well correlated between canine lymphoma cell lines (Figure 1). Studies reported that the expression of XPO1 in both gene and protein in the majority of canine osteosarcoma cells were significantly higher compared with normal canine osteoblast cells [14], and the XPO1 protein overexpression demonstrated potent activity necessary for cell survival in canine cancers [5]. In addition, XPO1 was also overexpressed compared to PBMC, normal B and T cells at the protein, as well as mRNA level in a human lymphoma/leukemia study [21,22,23]. In those reports, the authors did not compare the expression levels of XPO1 between mRNA and protein. The overexpression of XPO1 was also reported in other human cancers such as thymic epithelial tumors, malignant melanoma, osteosarcoma, glioma, ovarian, pancreatic, cervical, colorectal, and gastric cancers [7,24,25,26,27]. Our analysis showed that the XPO1 protein expression tended to be high in all the canine lymphoma cell lines examined, suggesting that they could serve as potential therapeutic targets in canine lymphoma cells.

Studies in human hematologic malignancies have demonstrated alterations in the aberrant expression of XPO1 including its overexpression, deregulation, and dysfunction. They showed several mechanisms such as amplification of the *XPO1* gene, chromosomal translocation with *XPO1* gene locus, and *XPO1* mutation as abnormalities [28,29,30]. Large-scale genomic analyses of human cancers identified mutational hotspots in the *XPO1* gene and also recurrent heterozygous *XPO1* mutations in a variety of cancer types, including chronic lymphocytic leukemia (CLL), Hodgkin’s lymphoma, and esophageal carcinoma [31]. These mutations resulted in the dysregulation of TSPs export, leading to tumorigenesis, aggravated disease progression, and poor survival outcomes [30,32]. Additionally, studies in B-cell lymphoma and non-small cell lung cancer (NSCLC) indicated that *XPO1* mutations may result in resistance to conventional chemotherapy; however, cells with mutated *XPO1* demonstrated increased sensitivity to XPO1 inhibitors so that the therapeutic remained effective [28,29]. We conducted *XPO1* gene mutation analysis in the eight canine lymphoma cell lines and found that there were no mutations presented in the coding region of the canine *XPO1* gene. Therefore, the cause of overexpression of XPO1 in the canine lymphoma cell lines observed in this study needs to be clarified in future studies.

Recently, studies using SINE compounds have been conducted in human hematologic malignancies. The efficacy of various types of SINE compounds was investigated in non-Hodgkin lymphoma (NHL), specifically T-cell lymphoma, Mantle cell lymphoma, and DLBCL cell lines, which showed a significant decrease in cell proliferation at concentrations ranging in nano-molar concentration up to 500 nM [21,23]. Some in vitro and in vivo studies in other human cancers also revealed low IC_50_ of various SINE compounds, affected the nuclear retention of many TSPs, and restored their function to induce cell cycle arrest and cell proliferation inhibition, leading to apoptosis [12,24,25,26]. Similar to those studies, our study showed that KPT-335 is effective against canine lymphoma cell lines and exhibited a low IC_50_ range at concentrations of 89.8–418 nM (Figure 2 and Table 1). These low IC_50_ indicated that KPT-335 has a significant anti-proliferative effect against canine lymphoma cells. In canine cancers, KPT-335 or verdinexor, have been exhibited good biological activities in canine malignant melanoma, osteosarcoma, mammary, and transitional cell carcinoma (TCC) cell lines in vitro [13,14,33]. Since the mean KPT-335 Cmax was reported to be 278 ng/mL (0.628 μM) in the phase II canine lymphoma study [17], our in vitro results at a concentration of 0.5 μM, close to the Cmax, support the therapeutic efficacy and tolerability of KPT-335 in canine lymphoma cases.

Consistent with the effects of cell proliferation, KPT-335 exposure in escalating concentrations showed a cytotoxicity effect which significantly reduced cell viability in canine lymphoma cell lines (Figure 3). Similar findings have been reported in human CLL and gastric cancer studies, where exposure to SINEs has led to the nuclear retention of TSPs such as FOXO, IκB, and p53, leading to cell cycle arrest, reducing cell viability and promoting cell death [22,26,33]. Based on our findings, KPT-335 has shown cytotoxic effects in canine lymphoma cell lines which are characterized by decreased cell viability; however, the localization of TSPs in the nucleus which leads to cell cycle arrest and apoptosis has yet to be clarified.

Although the *XPO1* gene and protein expression showed a significant correlation, our study found no correlation between the sensitivity to KPT-335 and the XPO1 mRNA and protein expression in canine lymphoma cell lines. Similar to our results, a study using canine osteosarcoma cells reported that the expression level of XPO1 protein was not directly correlated with sensitivity to KPT-335 [14]. Investigation in human hematologic tumor cell lines revealed that the capability of SINE compounds binding to XPO1 was similar, regardless of their sensitivity to the drugs [34]. No correlation was observed between drug sensitivity and dose-dependent occupancy, suggesting that sensitivity to KPT-335 may not depend solely on binding to XPO1, but also on XPO1’s target molecules. The observed lack of correlation points to additional intracellular determinants, such as apoptotic threshold and basal cell cycle activity, may influence therapeutic efficacy independently of XPO1 expression levels. Rapidly proliferating cells exhibit an increased dependency on nuclear export mechanisms, making them more susceptible to XPO1 inhibition. In human cutaneous T-cell lymphoma, the downregulation of survivin, a key inhibitor of apoptosis, following XPO1 inhibition may lower the cellular threshold for apoptosis, enhancing the drug sensitivity. The study also observed cell cycle arrest at G1 following treatment, suggesting that basal activity influences response dynamics [35]. In canine lymphoma cells, inactivation of p16 (CDKN2A) and hyperphosphorylation of its target, pRb, have been reported [18,36,37]. Our previous studies have shown that among the cell lines used in this study, CLBL-1, CLC, CLGL-90, Ema, Nody-1, and UL-1 exhibit hyperphosphorylation of pRb and/or inactivation of p16, whereas no aberrations in pRb or p16 were found in 17-71 and GL-1 [18]. Unfortunately, our results showed no significant difference in the IC_50_ values of KPT-335 between 17-71 and GL-1 and the other cell lines, suggesting that differences in molecular mechanisms beyond the p16-pRb pathway may influence sensitivity to KPT-335. Importantly, at least at this stage, the expression levels of XPO1 mRNA or protein do not appear to serve as reliable predictors of KPT-335 sensitivity in canine lymphoma cells.

An immunophenotyping study with canine lymphomas reported that 68.7% of canine lymphomas originate from B cells, while 31.3% are of T-cell origin [3], and the latter are less responsive to conventional treatments compared to B-cell lymphomas [1,3,6]. Based on the phase II study [17], we initially hypothesized that KPT-335 would be particularly effective against T-cell lymphomas, leading to the initiation of this in vitro study. However, no difference was observed in sensitivity between B-cell and T-cell lines. Given the wide heterogeneity of lymphoma subtypes, interpretation of these results remains challenging. To clarify the differences in KPT-335 sensitivity between B and T cells, it will be necessary to conduct clinical studies using a larger cohort over extended period.

Combination regimens incorporating XPO1 inhibitors with other anti-cancer agents have demonstrated synergistic potential across multiple human malignancies [12]. In human cancers, promising efficacy has been reported using XPO1 inhibitors in combination with standard chemotherapies, proteasome, or tyrosine kinase inhibitor [7,9,10,11]. Especially, clinical trials using XPO1 inhibitors in hematologic malignancies are being investigated in combination with rituximab, cyclophosphamide, doxorubicin, vincristine, and prednisone (R-CHOP) protocol, as well as targeted agents such as ibrutinib, or has been combined with corticosteroid and/or bortezomib [11,20]. In canine oncology, an in vitro study of osteosarcoma cell lines revealed enhanced cytotoxicity upon co-treatment with KPT-335 and anthracycline and may produce synergistic therapeutic potential [14]. Prior research involving SINE compounds in both human and canine malignancies supports the potential for combinatorial treatment utilizing KPT-335 and cytotoxic agents in clinical cases where resistance to conventional chemotherapy has emerged. Therefore, further studies are warranted to evaluate the efficacy of combining XPO1 inhibitors with other anticancer agents in the treatment of canine lymphoma.

This study demonstrated that KPT-335 effectively inhibited cell growth and reduced cell viability. However, this study has some limitations. First, further investigations were needed to provide advanced information about the effects of post-treatment of KPT-335 against canine lymphoma cell lines and the potential mechanism of KPT-335 in nuclear-cytoplasmic translocations of cargo proteins. Subsequently, samples from clinical cases should be used in future studies to evaluate the biological activity and the effectiveness of KPT-335.

## 5. Conclusions

The XPO1 mRNA and protein expressions in the canine lymphoma cell lines tended to be higher than those in PBMC as normal control, and the XPO1 mRNA expression was well correlated with the XPO1 protein expression. KPT-335 inhibits cell proliferation effectively and induces cell death in the canine lymphoma cell lines at nanomolar level. However, the XPO1 mRNA and protein expression were not associated with the efficacy of KPT-335. Overall, this study suggested that XPO1 may be a promising target for the treatment of canine lymphoma. Since drug resistance of traditional chemotherapy has occurred in canine lymphoma case reports, further investigations of drug synergism of XPO1 inhibitor with other anti-cancer drugs are needed.

## Figures and Tables

**Figure 1 vetsci-12-00700-f001:**
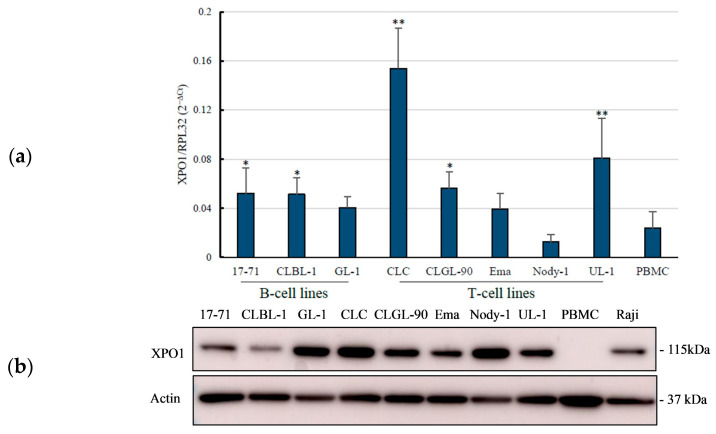
The *XPO1* gene and protein expressions in canine lymphoma cell lines. A quantitative real-time PCR was performed on eight canine lymphoma cell lines. (**a**) The Ct values of the *XPO1* gene are expressed relative to the *RPL32* gene. The XPO1 protein expression was assessed using the western blot analysis (**b**) and quantified using ImageJ (**c**). Protein expression of β-actin was used as an endogenous control. The data in the panel were presented as mean ± standard deviation (SD) from three independent experiments. *p*-values represented the significant differences of *XPO1* gene and protein expressions in canine lymphoma cell lines compared to those of PBMCs obtained from healthy dogs. * *p* < 0.05; ** *p* < 0.01.

**Figure 2 vetsci-12-00700-f002:**
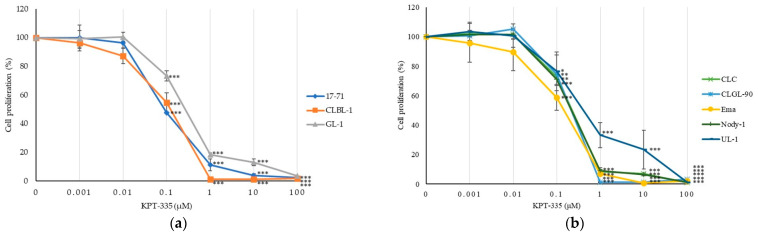
Inhibition of cell proliferation by KPT-335. Canine lymphoma cell lines, B-cell lines (**a**) and T-cell lines (**b**), were treated with increasing concentrations of KPT-335 for 48 h. Relative proliferation rates are shown as a percentage of the value of 0 μM as a control. The experiments were performed in triplicate and repeated three times. Error bars indicate SD. *p*-values indicated statistical differences between untreated and treated cells. * *p* < 0.05; ** *p* < 0.01; *** *p* < 0.001.

**Figure 3 vetsci-12-00700-f003:**
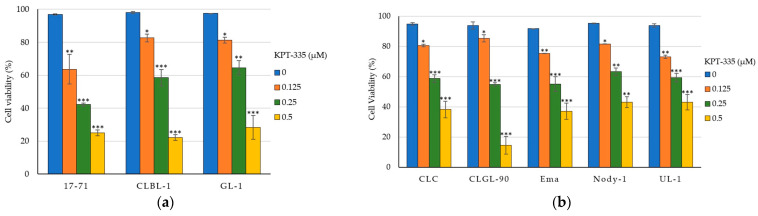
Decreased cell viability by KPT-335. KPT-335 was administered to canine lymphoma cell lines in (**a**) B-cell lines and (**b**) T-cell lines, with escalating concentrations (0, 0.125, 0.25, and 0.5 μM) for 48 h. The cell viability was determined by the percentage of living cells with the trypan blue exclusion method and compared between cells treated with and without KPT-335. The graphs are averages of triplicate wells from three independent experiments, with error bars representing SD. * *p* < 0.05; ** *p* < 0.01; *** *p* < 0.001.

**Table 1 vetsci-12-00700-t001:** IC_50_ concentrations for all canine lymphoma cell lines with KPT-335 treatment.

Cell Group	Cell Lines	KPT-335 IC_50_ (nM ± SD)
B-cell lines	17-71	89.8 ± 2.07
	CLBL-1	108.6 ± 30.54
	GL-1	294.3 ± 25.57
T-cell lines	CLC	224.7 ± 78.38
	CLGL-90	215 ± 50.96
	Ema	147.8 ± 49.69
	Nody-1	220.5 ± 27.34
	UL-1	418 ± 78.6

## Data Availability

The data presented in this study are included in the article. Further inquiries can be directed to the corresponding author.

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
