# Peer review of "Exportin 1 (XPO1) Expression and Effectiveness of XPO1 Inhibitor Against Canine Lymphoma Cell Lines"

_vetsci, 2025, doi:10.3390/vetsci12080700_

Round 1
Reviewer 1 Report
Comments and Suggestions for Authors
This manuscript presents an interesting investigation into XPO1 expression and effectivity of XPO1 inhibitor against canine lymphoma cell lines. The research is of considerable importance, contributing to our understanding of XPO1 expression and XPO1 inhibitor in canine lymphoma cell lines, as well as broader issues related to canine tumors and overall health. The manuscript is generally well-written, demonstrating a strong command of English, a logical structure, acceptable experimental design and data analysis, and appropriate use of current references. The findings of this study provide an effective target and a promising treatment of hematopoietic tumors in dogs. With attention to the minor issues outlined below, I recommend this manuscript for publication.
To further strengthen the manuscript, I suggest the authors consider:
1. The “Simple Summary” section lacks the intended simplicity and exhibits substantial overlap with the Abstract. Rephrasing of this section is therefore recommended.
2. A number of abbreviated annotations are repeatedly utilised, for example, XPO1 and TSPs. Conversely, several abbreviations lack annotation, such as HSD and WST.
3. The keywords could be further expanded to include terms such as cell lines, in vitro study, immunomodulation, and dogs, among others.
4. Care should be taken to ensure consistent and correct formatting of p-values throughout the manuscript. The authors are advised to consult the journal’s guidelines for specific formatting requirements.
5. Further formatting inconsistencies require attention throughout the manuscript, e.g., CO2 (L115), ver. (L165), etc.
Author Response
First of all, we deeply appreciate your helpful comments. Thank you very much for pointing this out. We agreed with your recommendation. Kindly refer to the attached file for the corrections.

Reviewer 2 Report
Comments and Suggestions for Authors
Summary of the Manuscript
This study investigates the expression of Exportin 1 (XPO1) at both the mRNA and protein levels in eight established canine lymphoma cell lines (three B-cell and five T-cell derived), and explores the in vitro efficacy of the XPO1 inhibitor KPT-335 (verdinexor). The authors confirm overexpression of XPO1 in five of eight cell lines using qRT-PCR and Western blotting. Furthermore, they show that all cell lines exhibit dose-dependent inhibition of cell proliferation and reduced viability in response to KPT-335, with IC50 values in the nanomolar range. Interestingly, the study finds a correlation between mRNA and protein expression of XPO1 but no clear correlation between XPO1 expression levels and sensitivity to the inhibitor.
The work is timely and methodologically sound, contributing to the growing interest in nuclear export inhibitors as novel therapeutic agents for canine lymphoma. The findings support further investigation into XPO1 as a molecular target, while also suggesting that predictive biomarkers for KPT-335 sensitivity may involve additional pathways beyond XPO1 expression alone.
Strengths and Positive Aspects
Scientific Relevance and Novelty
The study adds valuable data on the relationship between XPO1 expression and inhibitor sensitivity in canine lymphoma, an area with translational relevance to human oncology.
It builds upon previous preliminary in vivo findings and advances knowledge with comprehensive in vitro validation.
Methodological accuracy
The experimental design is robust, with well-described procedures for qRT-PCR, Western blotting, cell proliferation assays, and viability analysis.
Triplicate experiments and appropriate statistical tests (ANOVA, Kruskal-Wallis, Dunnett’s) increase confidence in the reliability of the results.
Clear Data Presentation
Figures are well-annotated and correspond clearly to the experimental narrative.
The inclusion of annotated Western blot images in the supplementary file enhances transparency.
Balanced Discussion
The manuscript acknowledges the limitations of correlating XPO1 levels with drug sensitivity and discusses possible mechanistic explanations including the p16/pRb pathway and prior findings from osteosarcoma and human lymphoma studies.
Language and Structure
The manuscript is clearly written with appropriate use of terminology and logical flow between sections.
Suggestions for Improvement
Clarify Cell Line Characteristics
It would improve interpretation if besides the immunophenotypic, additional molecular background of the eight cell lines (e.g., p53 and NF-κB status) were provided in a supplementary table.
Include whether any of the lines exhibit MDR/ABCB1 (multi-drug resistance) features.
Expand on XPO1 Functional Role
Given the lack of correlation between XPO1 expression and IC50, the authors might speculate more deeply about other determinants (e.g., efflux pumps, apoptotic threshold, basal cell cycle activity). This is somewhat left un-mentioned in the end of the discussion.
Clinical Translation
The discussion could benefit from more emphasis on potential clinical applications. For example: Could XPO1 expression be used as a screening tool in diagnostic biopsies? Is there scope for combination therapy with standard chemotherapeutics?
Minor Points
Some figures (especially Figure 2) could be improved in resolution for clarity of data points.
The conclusion mentions future exploration of drug synergy — this could be supported with references to known synergistic partners of XPO1 inhibitors in other species (e.g., corticosteroids, proteasome inhibitors).
Author Response
First of all, we really appreciate your thorough and detailed review. Thank you very much for the suggestion. Kindly refer to the attached file for the corrections.

Reviewer 3 Report
Comments and Suggestions for Authors
TITLE: In Vitro Analysis of Exportin 1 (XPO1) Expression and Effectivity of XPO1 Inhibitor Against Canine Lymphoma Cell Lines
1) Brief Summary
This work aimed to "investigate whether the sensitivity to KPT-335 is related to XPO1 gene and/or protein expression and investigate potential differences in drug sensitivity between B-cell and T-cell lymphoma cell lines."
To achieve their goals, the authors used eight established canine lymphoma cell lines." Isolated mRNA was subjected to quantitative real-time polymerase chain reaction using specific primers for Exportin 1 (XPO1), while the expression of XPO1 protein was evaluated by western blotting. The authors also assessed the efficacy of the XPO1 inhibitor (KPT-100) in canine lymphoma cell lines using cell proliferation and Trypan Blue exclusion assays.
The experiment has scientific merit, and the results are relevant. Still, I have a few issues and suggestions to further enhance the manuscript's merit.
2) General concept comments
I. General manuscript formatting
In general, the manuscript was formatted according to the template instructions.
II. Title: My suggested title is: "Exportin 1 (XPO1) Expression and Effectiveness of XPO1 Inhibitor Against Canine Lymphoma Cell Lines."
III. Simple Summary
According to the Microsoft Word Template available in the Instructions for Authors (https://www.mdpi.com/journal/vetsci/instructions), the simple summary "should be written for a lay audience, i.e., no technical terms without explanations. No references are cited, and no abbreviations."
In this case, the simple summary is inadequate. There are many technical terms and abbreviations. It is just a version of the Abstract. It must be reviewed following the parameters.
IV. Abstract
The Abstract is appropriate.
V. Introduction
The Introduction is clear and appropriate. My only suggestion for the Introduction is that the last sentence, " In the context of veterinary oncology, it will contribute good implications for future research and clinical applications in canine lymphoma" should be changed to "In the context of veterinary oncology, this work will provide auspicious implications for future research and clinical applications in canine lymphoma".
VI. Materials and Methods
This section is concise and mostly adequate. However, I have some concerns regarding this section for the authors.
The main issue is that the authors performed qPCR using various lymphoma lines and employed an internal reference (housekeeper) gene (RPL32), as described by Peters et al. (2007). However, it would be valuable to have data on the expression of XPO1 in regular B and T lymphocytes to establish a comparison. This strategy was employed by Breitbach et al. (2021), as referenced in #17 of the manuscript. By using cultured osteoblasts to compare the expression of XPO1 in osteosarcoma tissue samples and cell lines (Figure 1 of their publication) via quantitative PCR (qPCR) and Western Blotting, they effectively demonstrated that XPO1 is overexpressed in tumoral tissue and cells compared to normal cells.
It would be valuable to include the references for the primers used. It is implied in the text that the primers for RPL32 were used by Peters et al. (2007), but the reference is not apparent. Regarding the XPO1 primers, no information is available in the text.
VII. Results
Although the results are interesting, I have a few issues and questions for the authors.
a) The headings for 3.1 and 3.2 mention "overexpression of XPO1". However, in the description of the results, the authors (correctly) describe the expression of XPO1 as "increased". The term "overexpression" is, in my opinion, appropriate when compared to standard cell lines like PBMC, as used for the WB.
b) Equally, in lines 246-248, the authors report that "The XPO1 gene expressions in eight canine lymphoma cell lines were generally high and related to their XPO1 protein expression, except for CLBL-1, where the XPO1 protein expression analysis yielded no significant difference." It is implied that the RNAm expression is high compared to the internal control (RPL32).
c) Although the differences between cell lines are visible in the Western Blotting (WB) image, the pixels appear oversaturated. If the authors could produce a less exposed image, the contrast would be more evident.
d) I have a question for the authors. Why doesn't the WB image (Figures 1b and 1c show the controls (as seen in the supplementary material) for the sake of comparison?
e) Additionally, in the supplementary material, a band for XPO1 WB is not visible. In the "Western blotting images with annotation" membrane, a (-) sign is placed over a bar labeled "Control", possibly indicating that PBMC were used as a negative control. However, standard PBMC express XPO1 as seen, for example, in https://doi.org/10.1007/s10637-022-01250-6. The question is: have PBMC been misused as a negative control, or was the WB sensitivity insufficient to detect the XPO1 band?
VIII. Discussion
I also have considerations regarding the Discussion.
a) In the first two paragraphs of the Discussion, the authors compare their findings with those published by Breitbach et al (2021). However, as previously discussed in this review (VII. Results, a and b), the referred work demonstrated increased XPO1 expression for both RNAm and protein, compared to normal osteoblasts used as a control.
b) Likewise, the authors also refer to other reports (lines 304-305, references 21, 22, and 23) that "XPO1 was also overexpressed compared to normal B and T cells at the protein as well as mRNA level in a human lymphoma/leukemia study."
c) In the closing argument of the second paragraph, (lines 309-311), the authors claim that "Our analysis showed that the XPO1 protein expression tended to be high in all the canine lymphoma cell lines examined, suggesting that they could serve as potential therapeutic targets in canine lymphoma cells." In this case, if the lymphoma cell lines XPO1 bands in the Western Blotting is compared to PBMC presented in the Supplementary Material, the term "high expression" is appropriate. However, as shown in Figure 1, the expressions are only comparable among themselves, i.e., lymphoma cell lines. Furthermore, when RNAm expression is considered, the lack of a control with normal cells is restrictive in terms of using the term "overexpression".
d) A question for the authors: What are the "good biological activities" (line 340) of KPT-335 referred to by the authors?
IX. Conclusion
I also have questions and issues related to the Conclusions.
a) If the authors mean that their conclusions are based on results that will be obtained in a similar pattern if their experiments are repeated, they should rewrite the conclusions using the present tense. Example:
"KPT-335 inhibited cell proliferation effectively and induced cell death in the canine lymphoma cell lines at nanomolar level" (lines 392-393) should be: KPT-335 inhibits cell proliferation effectively and induces cell death in the canine lymphoma cell lines at nanomolar level".
b) At the beginning of the conclusion, the authors claimed that "The XPO1 mRNA and protein expressions in the canine lymphoma cell lines were (sic) tended to be higher". My question is: higher than which expression? From which cells? Please explain.
c) The authors also claimed (lines 393-395): "However, the XPO1 mRNA and protein expression were not associated with the efficacy of KPT-335. Overall, this study suggested that XPO1 may be a promising target for the treatment of canine lymphoma." My point is: how can this particular study, considering all the results ("overall", in the author's words), suggest that XPO1 is a promising target for treatment if the favorable results of the treatment protocol used in the second set of experiments did not correlate with XPO1 expression? This conclusion would be acceptable if only the results from the XPO1 expression were considered, particularly if expression controls (standard lymphocytes) had been used.
Author Response
First of all, we would like to express our deep appreciation for the comprehensive summary and beneficial comments and suggestions. The manuscript has been revised as reviewer suggested. Kindly refer to the attached file for the corrections.

Round 2
Reviewer 3 Report
Comments and Suggestions for Authors
Thank you for updating your manuscript, improving the quality and relevance of this work.